# Development, Investigation, and Comparative Study of the Effects of Various Metal Oxides on Optical Electrochemical Properties Using a Doped PANI Matrix

**DOI:** 10.3390/polym13193344

**Published:** 2021-09-29

**Authors:** Amina Bekhoukh, Imane Moulefera, Lilia Sabantina, Abdelghani Benyoucef

**Affiliations:** 1L.M.A.E. Laboratory, Faculty of Science and Technology, University of Mustapha Stambouli Mascara, Mascara 29000, Algeria; amina.bekhoukh@univ-mascara.dz (A.B.); imanemoulefera@yahoo.fr (I.M.); 2Department of Chemical Engineering, Faculty of Science, University of Málaga, Andalucía TECH, 29071 Málaga, Spain; 3Junior Research Group “Nanomaterials”, Faculty of Engineering and Mathematics, Bielefeld University of Applied Sciences, 33619 Bielefeld, Germany; lilia.sabantina@fh-bielefeld.de

**Keywords:** polyaniline, aluminum oxide, titanium dioxide, titanium carbide, nanocomposites, electrochemical properties

## Abstract

A comparative study was performed in order to analyze the effect of metal oxide (MO) on the properties of a polymeric matrix. In this study, polyaniline (PANI)@Al_2_O_3_, PANI@TiC, and PANI@TiO_2_ nanocomposites were synthesized using in situ polymerization with ammonium persulfate as an oxidant. The prepared materials were characterized by various analytical methods such as X-ray photoelectron spectroscopy (XPS), X-ray diffraction (XRD), thermogravimetric analysis (TGA), UV/visible (UV/Vis) spectroscopy, Fourier-transform infrared spectroscopy (FTIR), and transmission electron microscopy (TEM). Furthermore, the conductive properties of the materials were tested using the four-point probe method. The presence of MO in the final product was confirmed by XPS, XRD, FTIR, and TEM, while spectroscopic characterization revealed interactions between the MOs and PANI. The results showed that the thermal stability was improved when the MO was incorporated into the polymeric matrix. Moreover, the results revealed that incorporating TiO_2_ into the PANI matrix improves the optical bandgap of the nanocomposite and decreases electrical conductivity compared to other conducting materials. Furthermore, the electrochemical properties of the hybrid nanocomposites were tested by cyclic voltammetry (CV) and galvanostatic charge/discharge (GCD). The obtained results suggest that the PANI@TiO_2_ nanocomposite could be a promising electrode material candidate for high-performance supercapacitor applications.

## 1. Introduction

Recently, research on hybrid materials, which mainly focuses on conducting polymers with nanoparticles such as metal oxide (MOs) and Al_2_O_3_, TiO_2_, and TiC, has attracted considerable attention due to their wide application in various fields such as sensors, electrodes, batteries, photovoltaics, medicine, electrochemistry, and energy, as well as their ease of fabrication [1]. Additionally, conducting polymers have earned an important role as they are low-cost, are environmentally friendly, and have low toxicity [2,3]. Among them, polyaniline (PANI) is a conducting polymer that has received considerable attention in the industry due to its environmental and thermal stability [4,5,6]. Moreover, due to its properties such as high conductivity, broad absorbance, simple synthesis, ease of coupling with metal ions, porous morphology structure, and environmental stability, PANI can be used in numerous applications such as supercapacitors, electrochromic devices, sensors, light-emitting diodes (LEDs), and anti-corrosion coatings [7,8,9,10,11,12,13].

Due to the large potential of conducting polymers and polymers doped with inorganic nanoparticles, which improve the performance of materials in many applications, the fabrication of nanocomposites seems to be a recent challenge. Many authors studied the synthesis of nanocomposites of polyaniline with various nanoparticles such as Fe_3_O_4_ [14], ZnO [15], ZrO_2_ [16], TiO_2_ [17], Al_2_O_3_ [18], TiC [4], and montmorillonite [19]. With the doping of conductive polymers with metal oxides, defined properties such as high electrical conductivity, high electron affinity, and improved mechanical properties can be achieved [20]. Such conductive polymers doped with nanomaterials are used for different purposes and applications. Naveed-ur-Rehman et al. proposed conducting polymers with metal oxide nanoparticles for the production of electrode materials for supercapacitors. In their study, various nanocomposites containing Co_3_O_4_, Pr_2_O_3_, and NiO nanoparticles, binary PANI–Pr_2_O_3_, PANI–NiO, and PANI–Co_3_O_4_ nanoparticles, ternary Pr_2_O_3_–NiO–Co_3_O_4_ nanoparticles, and quaternary PANI–Pr_2_O_3_–NiO–Co_3_O_4_ nanoparticles with a spherical core and shell were prepared using coprecipitation and ultrasonic methods [21]. The review by Mohd Abdah et al. explored the use of fibers based on transition metal oxides and conducting polymers for the application area of high-performance supercapacitors [22]. PANI and its composite with MnO_2_ were successfully deposited by electrochemical methods in the study by Rabbani et al., and the composite electrodes with Fe-PANI and Fe-PANI/MnO_2_ showed excellent supercapacitive properties [23]. Lokhanni et al. prepared supercapacitive metal oxide composite electrodes composed of carbon, metal oxides, and conducting polymers [24].

The preparation of nanocomposites with high performance such as high thermal stability and good electrochemical properties has become a major challenge according to the dispersion of nanoparticles in the polymers [25]. To increase the dispersion of nanofillers, the commix method and stearic acid are used to prevent the aggregation of nanoparticles in the polymer matrix [26].

MOs exhibit different electrical behavior that can change from electrically insulating to others such as Al_2_O_3_ and MgO, as well as wide-band semiconductors such as SnO_2_, TiO_2_, Ti_2_O_3_, and ZnO, and metal-like behaviors such as ReO_3_, V_2_O_3_, and RuO_2_. The metal oxide presents several stable oxidation states, which play a significant role in the surface chemistry of the obtained copolymer [27].

The need for new materials with promising electrochemical performances represents an important target for energy storage applications. In addition, the development of new-generation supercapacitors is associated with numerous problems, such as high self-charge/discharge currents, low energy density, high equivalent series resistance, and low operating potential. Although MO electrodes are available for supercapacitors, they suffer from poor cycling stability and low current capability [26]. In addition, MOs are relatively expensive compared to carbon materials. Therefore, there is a great need to improve the electrochemical properties of MOs and CPs.

In the present work, three different nanocomposites were prepared from polyaniline and nanoparticles (Al_2_O_3_, TiC, and TiO_2_) using the in situ polymerization method in the presence of HCl with ammonium persulfate as an oxidant. The obtained samples were characterized by XRD, XPS, TGA, FTIR, TEM, and CV in order to determine the thermal stability and electrochemical behavior.

## 2. Materials and Methods

### 2.1. Materials

The monomer aniline (ANI) (purity ≥ 99.5%, Sigma-Aldrich, St. Louis, MI, USA), perchloric acid (HClO_4_, 70%, Merck KGaA, Darmstadt, Germany), hydrochloric acid (HCl, 37%, Merck KGaA, Darmstadt, Germany), and ultrapure H_2_O (18.2 MΩ cm, Elga-Labwater-Purelab system, ELGA LabWater, Veolia Water Technologies Deutschland GmbH, Celle, Germany) were used in all the experiments. Aluminum oxide nanoparticles (Al_2_O_3_) (≥99.98%, Merck KGaA, Darmstadt, Germany), titanium (IV) oxide (TiO_2_) (≥99.98%, Merck KGaA, Darmstadt, Germany), zinc oxide (ZnO) (99%, Merck KGaA, Darmstadt, Germany), ammonium persulfate (APS) (≥98%, Merck KGaA, Darmstadt, Germany), and ammonia solution (NH_4_OH) (25%, Merck KGaA, Darmstadt, Germany) were used for the preparation of nanocomposites.

### 2.2. Chemical Preparation of Hybrid Materials

PANI and its corresponding nanocomposites, PANI@Al_2_O_3_, PANI@TiO_2_, and PANI@TiC, were prepared via in situ oxidative polymerization reactions [4,20,21,22]. A defined amount of 1 g for each MOs (Al_2_O_3_, TiC, or TiO_2_) was dispersed in 1 M HCl solution under magnetic stirring for 30 min. Then, 135 mg of ANI monomers were added to this mixture dropwise and stirred for 30 min to inhibit nanoparticle agglomeration and allow for the electrostatic interactions to deposit HCl-doped aniline on the surface of MOs. Afterward, the polymerization was initiated by the APS added dropwise in a stoichiometric ratio to the monomer (applied 1:1 oxidant/monomers molar ratio). The mixture was stirred for 24 h at room temperature. The next day, the precipitate was collected on a filter paper, washed with distilled H_2_O and acetone, and finally placed in 50 mL of NH_4_OH (1 M) at 25 °C and stirred for 2 h. The final products were filtered, washed with deionized water, and dried under vacuum at 65 °C for 24 h [4,27,28,29].

### 2.3. Physicochemical Characterization

X-ray photoelectron spectroscopy (XPS) was performed to measure the surface components of materials using an AVG-Microtech-Multilab 3000-electron spectrometer (VG Microtech Ltd., London, UK). A Hitachi U-3000 spectrophotometer (Hitachi High-Technologies Corporation, Tokio, Japan) was used to measure UV/Vis spectra. The X-ray diffraction of the samples was acquired by applying a Bruker CCD-Apex (Madison, WI, USA) equipment. For transmission electron microscopy (TEM) observations, the materials were dispersed in ethanol and supported on TEM grids. The images were collected using a JEOL-JEM-2010 microscope (Jeol, Peabody, MA, USA). Thermogravimetric analysis (TGA) (Hitachi STA7200 instrument, Fukuoka, Japan) was performed with a DuPont thermogravimetric analyzer at a heating rate of 20 K·min^−1^ under N_2_. About 10 mg of material was heated up to 900 °C. Fourier-transform infrared (FTIR) spectroscopy (Varian, Inc., Palo Alto, CA, USA) was performed using the Bruker-Alpha equipment [28,29].

Electrical conductivity measurements were performed using a Lucas Lab resistivity equipment with four in-line probes. The materials were dried for 24 h, and then pellets with a diameter of 0.013 cm were prepared using an FTIR mold by applying a pressure of 7.4 × 10^8^ Pa.

### 2.4. Electrochemical Properties

The electrochemical behavior of the materials was characterized by cyclic voltammetry (CV). The materials were first dissolved in NMP [4]. Then, a drop of the resulting solution was placed on the graphite carbon electrode and dried in air under an infrared lamp to solvent remove. The electrochemical studies were performed using a conventional three-electrode cell. The reference and counter electrodes were a Pt wire and a reversible hydrogen electrode (RHE) submerged in the electrolyte, respectively. HClO_4_ (1 M) was used as an electrolyte in all experiments.

## 3. Results and Discussions

### 3.1. X-ray Photoelectron Spectroscopy (XPS) Spectra

The surface characteristics of PANI@Al_2_O_3_, PANI@TiO_2_, and PANI@TiC samples were investigated by XPS analyses. The XPS spectra of all hybrid materials showed the existence of contributions from N 1*s*, confirming the presence of the PANI chain in these samples (Figure 1). Furthermore, the results show that the addition of the MOs used as support in the polymer matrix led to the detection of new peaks. Therefore, PANI@TiO_2_ and PANI@TiC nanocomposites showed peaks at 459 eV and 462 eV, which correspond to Ti 2*p*_3/2_ and Ti 2*p*_1/2_, respectively. In addition, two new peaks near 74 eV and 117 eV were visible in the XPS spectrum of PANI@Al_2_O_3_, which were assigned to Al 2*p* and Al 2*s* species, respectively.

To further investigate the change in the electronic structure of the hybrid materials, Gaussian deconvolution peak integration was used to calculate the respective proportions of the N 1*s* peak (Figure 2). In this way, the N 1*s* spectrum of the PANI@TiO_2_ sample can be subdivided into three peaks, for which the assignment of the binding energies is listed in Table 1. The first peak at 398.39 eV corresponded to neutral amines, the second peak at 399.53 eV corresponded to imine species, and the last peak at 400.33 eV could be assigned to the nitrogen cationic radical (–N^+^), which is also present in the structure of PANI [30]. Similarly, for the PANI@TiC sample, the three peaks appearing at 398.51 eV, 399.71 eV, and 400.83 eV corresponded to quinoid amine (–N=) and benzenoid amine (–NH– and –N^+^). In contrast, the spectrum of PANI@Al_2_O_3_ showed only two peaks at 398.22 eV and 399.48 eV, which could be assigned to −N= and −NH−, respectively.

On the other hand, the ratio of the areas under [=N–]/[–NH–] compositions indicates the quantitative intrinsic oxidation state (OS) of the samples [23]. The area ratio of the lowest binding energy to the total area band is an indicator of the doping degree (DD) [30]. The quantitative data on the intrinsic oxidation state and doping level of the hybrid materials are listed in Table 1. The OS of PANI@TiC was 0.56, and the defect density was only 0.05, while the OS of PANI@TiO_2_ was 0.29, and the defect density was up to 0.51. This latter nanocomposite had a weak OS and higher DD, which probably resulted in a faster charging/discharging ratio and significant capacitance. These different data of the N 1*s* spectra indicate that these conducting materials had different distributions, i.e., different doping states.

### 3.2. Fourier-Transform Infrared Spectroscopy (FTIR) Spectra

Figure 3 shows the FTIR spectra of PANI and its nanocomposites. The characteristic peaks of PANI at 3088–3258 cm^−1^ were attributable to the stretching of N–H [31,32,33], and the peaks at 1643–1546 cm^−1^ and 1472–1506 cm^−1^ were due to the C=N and C=C stretching modes for the quinoid and benzenoid rings [32,33,34], respectively. The bands at about 1304 cm^−1^ and 1243 cm^−1^ corresponded to the stretching of C–N bonds in the benzenoid ring, whereas the bands between 1000 and 1129 cm^−1^ were assigned to a planar vibrational flexion of C–H bonds [32,33,34]. The band at 803 cm^−1^ represented the stretching of C–H out-of-plane bending vibration [31].

The FTIR spectra of nanocomposites show that these materials contained the same characteristic PANI bands, while some bands presented a shifting compared to PANI, which may be associated with the interaction existing between MO and polymer.

The spectrum of the PANI@TiO_2_ nanocomposite showed a characteristic band at 732 cm^−1^, which was attributed to the Ti–O stretching band [35]. The presence of TiC bands at 635 cm^−1^ and 550 cm^−1^ in the IR spectrum of PANI@TiC nanocomposite confirmed the incorporation of TiC into the PANI matrix [4]. For the PANI@Al_2_O_3_ nanocomposite, the band at 503 cm^−1^ was seen to be attributed to condensed octahedral AlO_6_ [18]. The shift of the peak positions observed confirmed the interaction between PANI and Al_2_O_3_.

### 3.3. X-ray Diffraction (XRD) Studies

Figure 4 exhibits a comparison of the XRD patterns of PANI, TiO_2_, PANI/TiO_2_, TiC, PANI@TiC, Al_2_O_3_, and PANI@Al_2_O_3_. The XRD pattern of PANI displayed two peaks at 2*θ* = 20° and 25°, which were attributed to the (100) and (110) planes, respectively [31,35]. These peaks can be attributed to the parallel and perpendicular periodicity of the PANI chain and are characteristic of the protonated form of PANI [35,36]. For nanocomposites, all peaks were present for the MOs of TiC, Al_2_O_3_, and TiO_2_, while the broad peaks between 10° and 30° attributed to the amorphous structures of PANI confirmed the polymer–MO interaction [18]. The characteristic diffraction patterns of the titania anatase phase (TiO_2_) were observed at 2*θ* = 25.28°, 36.9°, 37.81°, 38.57°, 48.05°, 53.85°, 55.03°, 62.12°, 62.69°, and 68.76° which were associated with the (101), (103), (004), (112), (200), (105), (211), (213), (204), and (116) crystal planes, respectively [37]. A peak around 32° was found in the PANI@TiO_2_ nanocomposites, which could be attributed to the polymers, and there was a small peak at an angular position of 2*θ* = 23°, which represented the emeraldine salt form of PANI.

### 3.4. UV/Visible (UV/Vis) Spectroscopy

Figure 5a presents the UV/Vis absorption spectra of PANI, PANI@TiC, PANI@TiO_2_, and PANI@Al_2_O_3_, where these samples were dissolved in NMP. The materials displayed two characteristic absorption bands. The first band at 314–394 nm could be attributed to the π–π* transition of the benzenoid ring; the second band between 591 and 616 nm could be ascribed to the transition of the exciton of the quinone, indicating the delocalization of electrons in the polymer [38] (the various transitions are included in Table 2).

In the nanocomposites, the intensity of the π–π* and n–π* transitions of the bands decreased compared to PANI, and these bands were shifted to lower wavelengths [11]. They affected the electron distribution in the benzene ring region and formed a large conjugated system, which was due to the incorporation of MOs into the PANI matrix [33]. This confirmed that the presence of nanoparticles had an impact on the level of PANI doping, which was evident in the crystalline form [33].

It can be observed that the PANI@TiO_2_ exhibited the lowest intensity, indicating that the doping level of the TiO_2_-based nanocomposite was lower than that of TiC- and Al_2_O_3_-based samples.

The results of the UV/Vis investigation showed an electronic interaction between MO and PANI, which was attributed to a coordination bond between the pair of lone electrons of amine nitrogen and the empty orbitals of MO [39].

The UV/Vis spectra provide insight into the optical properties and bandgap energy values. The indirect bandgap energies of materials can be obtained from the Tauc plot (Figure 5b) of (*A^2^*) versus photon energy (*hν*) as follows [30]:$$A={\left(hv-E_{g}\right)}^{1/2},$$
where *h* is Planck’s constant, *h**𝜈* is photon energy, *A* is the absorption coefficient for direct transitions, *n = ½*, and *E_g_* is the optical energy gap.

The bandgap values obtained were 3.22 eV, 3.20 eV, 2.64 eV, and 2.62 eV for PANI, PANI@TiC, PANI@Al_2_O_3_, and PANI@TiO_2_, respectively (Table 2). Furthermore, the bandgap energy value of the PANI@TiO_2_ sample was lower compared to the other cases due to the strong interaction between PANI and TiO_2_, which caused changes in the electron density of the PANI chain. These changes caused a red shift, i.e., the absorption shifted to a longer wavelength [40]. On the other hand, this observed trend could be explained by the different crystallite structures. In addition to the impact of the crystal phase, the size of the MOs is considered to play a dominant role in their optical properties.

### 3.5. Thermogravimetric Analysis (TGA)

Using TGA, the thermal stability of the synthesized materials was investigated by monitoring the reduction in mass as a function of temperature. The TGA profiles of all samples are shown in Figure 6. The curves of each hybrid material exhibited a common feature, i.e., thermal decomposition in three different ranges. An initial weight loss (~5%) at a temperature >120 °C was associated with the evaporation of entrapped water, solvent, or monomer molecules in the sample [41]. This was followed by a second weight loss in the range of 150 to 450 °C due to the PANI chain doping and polymer moiety degradation [18]. In the third stage, which occurred at a temperature of >550 °C, the thermal stability was different; the weight loss varied between 15% and 30% for hybrid materials, whereas it was 57% for PANI. These results indicate that the nanocomposite of PANI@Al_2_O_3_ exhibited higher resistance to thermal degradation, as it had a total mass loss of 15% at 700 °C. In contrast, the PANI@TiC and PANI@TiO_2_ nanocomposites exhibited a higher mass loss of 28% and 30%, respectively. Moreover, the TGA curve of PANI@Al_2_O_3_ displayed a lower weight loss compared to the other two nanocomposites, indicating an improved thermal stability. This pronounced enhancement in thermal stability was mainly due to the strong π-stacking interaction between the benzene rings from PANI (aniline molecules) and the Al_2_O_3_ nanoparticle surface [42]. Due to the large surface area of the nanostructured Al_2_O_3_, the contact area between PANI and Al_2_O_3_ was sufficiently large to enhance the stability. It has previously been demonstrated that nanoparticles of TiC, Al_2_O_3_, and TiO_2_ exhibit strong thermal stability in the studied temperature range [4,18,28,43].

### 3.6. Transmission Electron Microscopy (TEM) Analysis

Figure 7 shows the TEM images of Al_2_O_3_, TiC, and TiO_2_ nanoparticles and PANI@Al_2_O_3_, PANI@TiC, and PANI@TiO_2_ nanocomposites. The MO nanoparticles in Figure 7A,C depict a spherical morphology, except for the TiC nanoparticles Figure 7B, which had a cuboidal shape. The average size of the Al_2_O_3_ and TiO_2_ nanoparticles was 37 nm and 28 nm, respectively. Moreover, the PANI@Al_2_O_3_ nanocomposite surface (Figure 7D) became more distinct compared to the pure nanoparticles (Figure 7A), which could be attributed to polymerization at the Al_2_O_3_ surface and a strong interfacial interaction between the nanoparticles and the PANI matrix [42]. However, the TEM image of TiC nanoparticles displayed a black streak [43]. Moreover, TEM images of the nanocomposites show that all MOs were well dispersed in the PANI matrix [44]. In addition, the morphology of all hybrid materials was found to be spherical with a size of 34–82 nm.

### 3.7. Electrochemical Properties

Cyclic voltammetry (CV) was performed to examine the polymer electroactivity. Figure 8a displays the CVs for PANI, PANI@TiC, PANI@Al_2_O_3_, and PANI@TiO_2_ materials obtained in HClO_4_ solution (1 M) at a scan rate of 50 mV·s^−1^.

All samples exhibited a nearly rectangular shape with two sets of distinct redox activities, as indicated by the two pairs of anodic and cathodic current peaks [44]. This shape is characteristic for supercapacitors and the typical Faradaic energy storage mechanism of PANI, which has been reported several times in the literature [8]. Furthermore, the anodic and cathodic peaks of PANI@Al_2_O_3_ were observed to be symmetrical, reflecting superior reversibility of the relevant redox reactions, and that most of the energy is stored by Faradic reactions [45], with the first peaks occurring at 0.48 V/0.32 V (Table 2), resulting in a potential peak separation (∆*E_p_*) close to 160 mV. These peaks corresponded to the oxidation of the leucoemeraldine–emeraldine form. Another pair of peaks at 0.86 V/0.74 V presented a ∆*E_p_* value of 120 mV corresponding to the oxidation of the emeraldine to pernigraniline of polyaniline [38,40]. The presence of TiO_2_ nanoparticles in PANI@TiO_2_ resulted in a shift of ox/red peaks to a lower potential. This indicates that the reaction kinetics were surface-limited, and that the electrochemical properties were determined not only by the nature of the polymer matrix, but also by the MOs [46].

Figure 8b shows the galvanostatic charge/discharge (GCD) curves of all the materials, which were carried out at a current density of 1.0 A·g^−1^ in the potential range from 0.2 to 0.8 V. The charging slopes of PANI@TiO_2_ showed a fast voltage rise from 0 V to 0.3 V, which was also due to the equivalent series resistance (ESR), followed by a flattening of the curve at about 0.4 V to 0.5 V, due to the oxidation of emeraldine to pernigraniline, the major redox transition where energy is stored [7]. In contrast, PANI@Al_2_O_3_ exhibited a distorted triangular shape with a shoulder at 0.78 V. This distortion was caused by the pseudocapacitive behavior due to the fast reaction of the PANI redox processes. However, the charging and discharging time constants were much larger for the other samples, indicating an important contribution of the Al_2_O_3_ nanoparticles to the charge storage process. In addition, the PANI@TiC curve exhibited an almost equilateral triangle shape, also indicating the good efficiency of the GCD process. There is a linear relationship between potential and time during GCD processes, which is another reason for the capacitance behavior of a material in addition to exhibiting rectangular CVs. Therefore, it is clear that the presence of crystalline MO nanoparticles with high porosity could enhance the stability of the materials. This result confirmed that the nanocomposites had a high electrochemical reversibility.

Cycling stability is a key factor in the operation of supercapacitors. Conductive polymers in supercapacitors often have limited cycling stability due to shrinkage and swelling during charging/discharging operation [7,47]. In this work, the cycling performance of the hybrid materials exhibited excellent stability with capacity retention between 78.4% and 84.8% after 1000 cycles at a high current density of 1.0 A·g^−1^ (Figure 8c), whereas PANI@TiO_2_ showed superior cycling stability (84.8%) without significant loss of specific capacitance compared to the other samples.

### 3.8. Conductivity Measurement

Conductivity measurements were carried out by a four-point probe method. The conductivity of the hybrid materials was low compared to that of pure PANI. The main reason seemed to be stereochemical differences between these nanocomposites. The oxidized PANI had an almost planar structure with a low ionization potential due to strong delocalization of the electrons. The conductivity values of PANI, PANI@TiO_2_, PANI@Al_2_O_3_, and PANI@TiC were 1.457 S·cm^−1^, 1.029 S·cm^−1^, 0.921 S·cm^−1^, and 0.682 S·cm^−1^, respectively. Moreover, the CV and GCD were used to test the electrochemical behavior of the hybrid nanocomposites. The results indicate that the PANI@TiO_2_ nanocomposite is a potential candidate for use as an electroactive electrode material in electronic devices.

## 4. Conclusions

Three PANI@MO hybrid materials were synthesized via in situ polymerization using ammonium persulfate as an oxidant. For this purpose, a comparative study of three metal oxides (MOs) (TiO_2_, TiC, and Al_2_O_3_) was carried out. The as X-ray photoelectron spectroscopy (XPS), X-ray diffraction (XRD), and Fourier-transform infrared spectroscopy (FTIR) analyses confirmed the successful preparation of the PANI@MO nanocomposites. Furthermore, the transmission electron microscopy (TEM) images displayed the spherical morphology of nanocomposites with a size of 34–82 nm. In addition, the optical properties of the samples were investigated. The optical bandgap energies reached about 3.22 eV, 3.20 eV, 2.64 eV, and 2.62 eV for PANI, PANI@TiC, PANI@Al_2_O_3_, and PANI@TiO_2_ samples, respectively. The electrical conductivity of the organic@inorganic samples was lower than that of pure PANI. Moreover, the thermogravimetric analysis (TGA) results showed that the decomposition of the nanocomposites was lower than that of the pure polyaniline, confirming the successful synthesis of the products. Furthermore, the electrochemical properties of the materials were tested by cyclic voltammetry, galvanostatic charge/discharge, and cycling stability techniques. The findings of this work may significantly advance the future of supercapacitor electrodes made from nanocomposite materials.

## Figures and Tables

**Figure 1 polymers-13-03344-f001:**
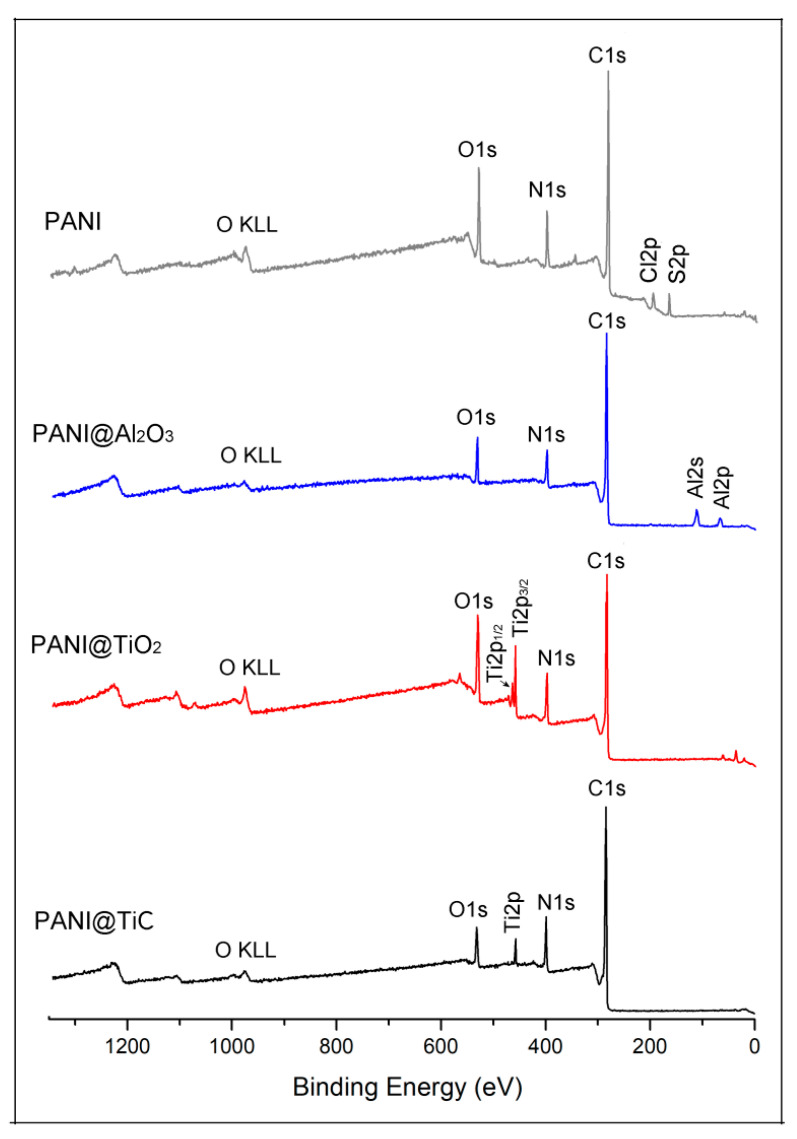
X-ray photoelectron spectroscopy (XPS) survey scan spectrum of synthesized materials.

**Figure 2 polymers-13-03344-f002:**
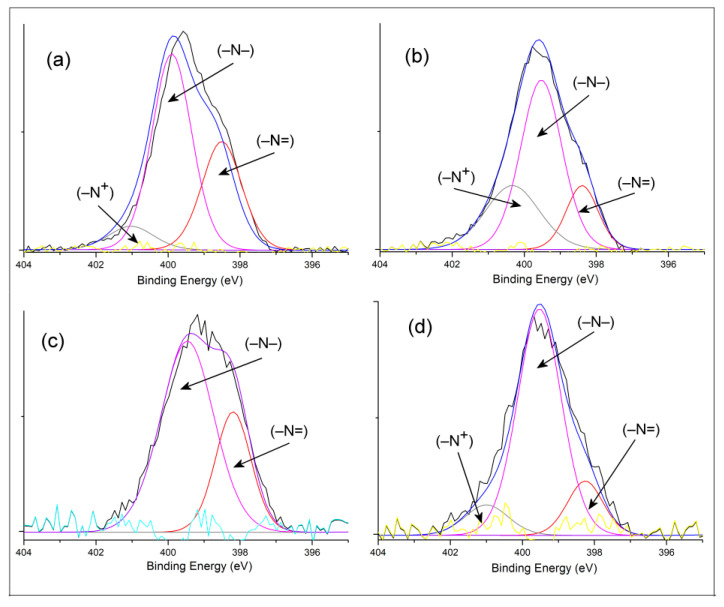
XPS spectra in N 1*s* region of (**a**) PANI@TiC, (**b**) PANI@TiO_2_, (**c**) PANI@Al_2_O_3_, and (**d**) PANI samples.

**Figure 3 polymers-13-03344-f003:**
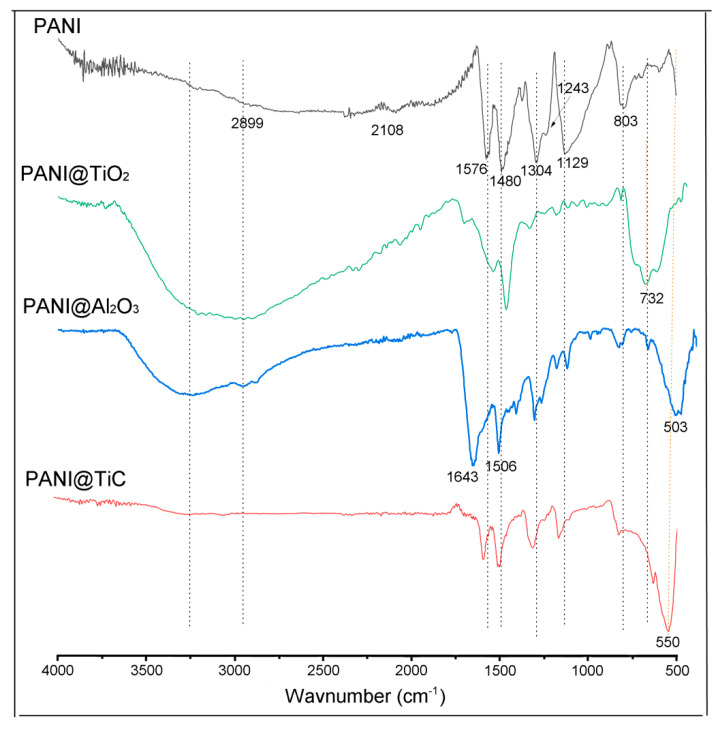
Fourier-transform infrared spectroscopy (FTIR) spectra of synthesized materials.

**Figure 4 polymers-13-03344-f004:**
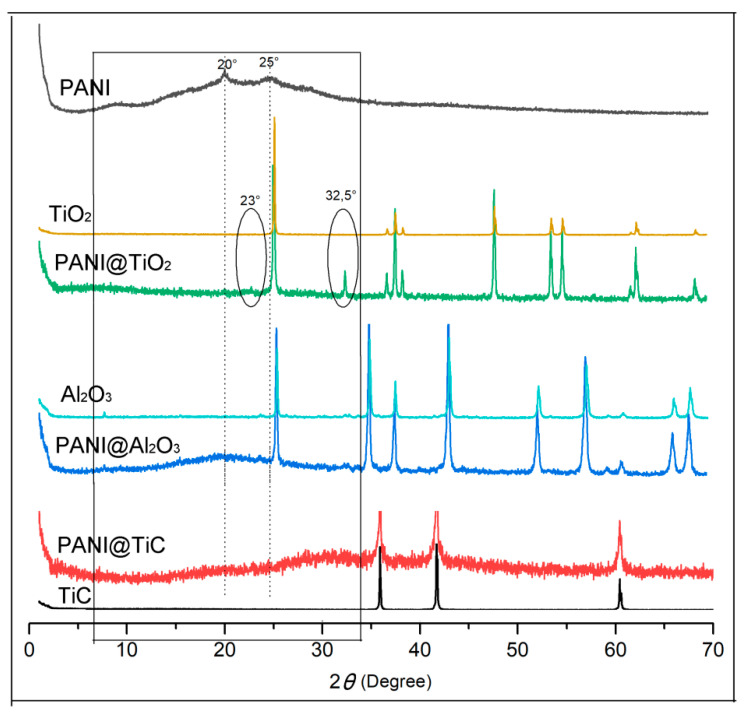
X-ray diffraction (XRD) patterns of the PANI, PANI@TiO_2_, PANI@TiC, PANI@Al_2_O_3_, TiO_2_, TiC, and Al_2_O_3_ materials.

**Figure 5 polymers-13-03344-f005:**
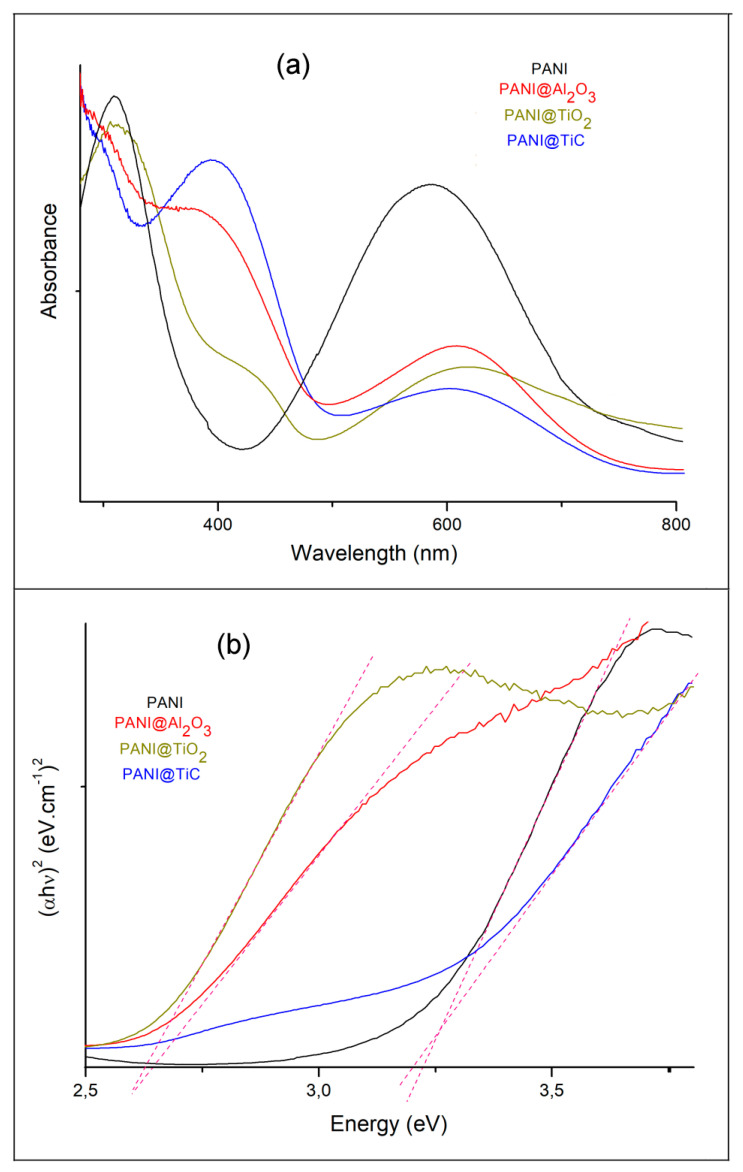
UV/visible (UV/Vis) spectroscopy absorption spectra (**a**) and Tauc plots (**b**) of synthesized samples.

**Figure 6 polymers-13-03344-f006:**
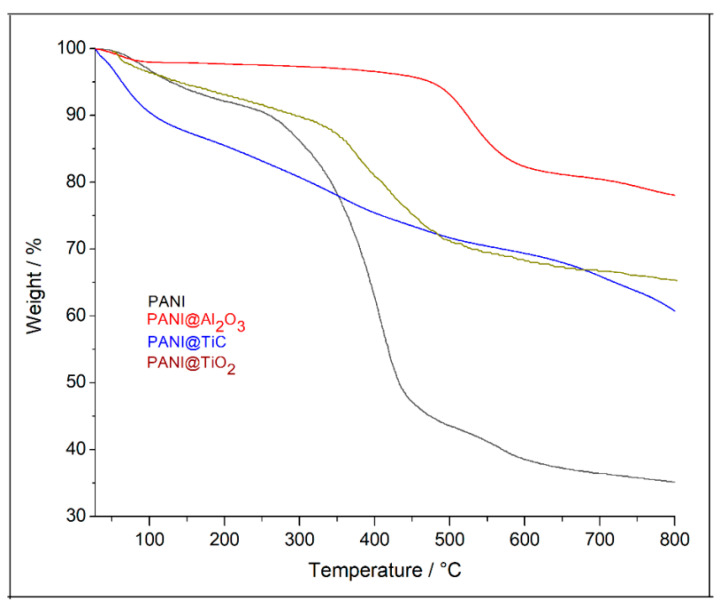
Thermogravimetric analysis (TGA) curves of PANI, PANI@Al_2_O_3_, PANI@TiO_2_, and PANI@TiC.

**Figure 7 polymers-13-03344-f007:**
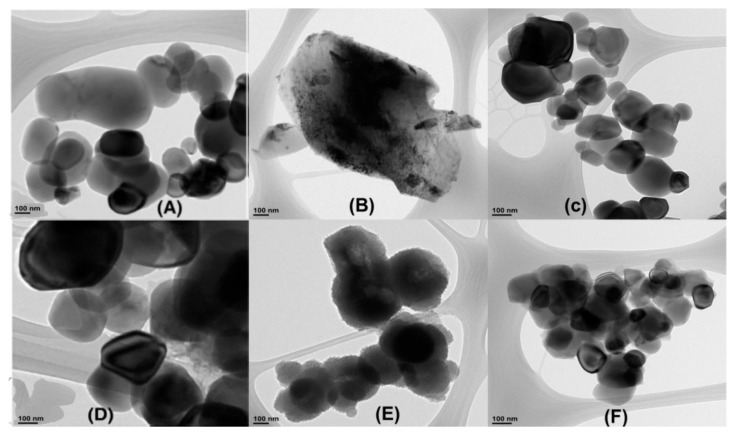
Transmission electron microscopy (TEM) images of (**A**) Al_2_O_3_, (**B**) TiC, (**C**) TiO_2_, (**D**) PANI@Al_2_O_3_, (**E**) PANI@TiC, and (**F**) PANI@TiO_2_.

**Figure 8 polymers-13-03344-f008:**
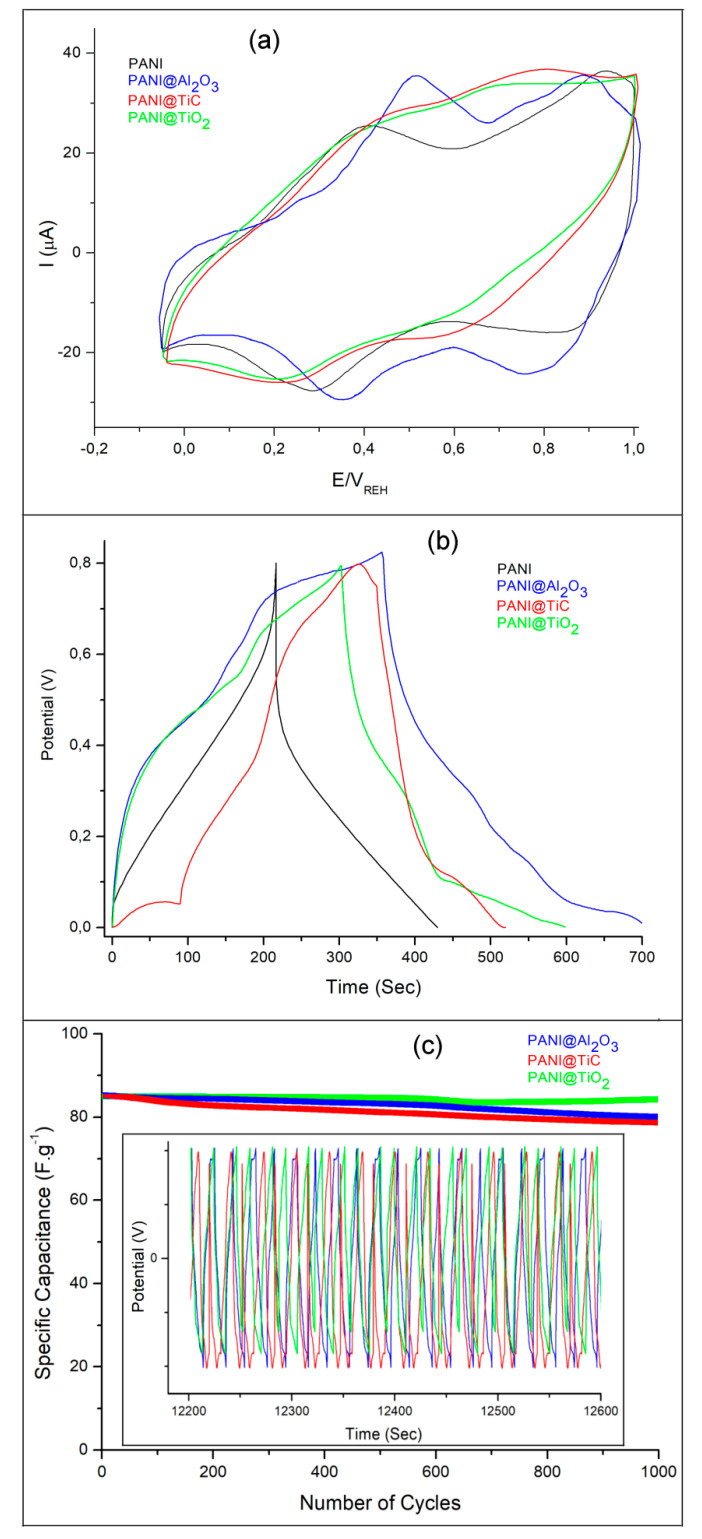
(**a**) Cyclic voltammograms recorded for a graphite carbon electrode covered by samples in HClO_4_ solution (1 M) at a scan rate of 50 mV·s^−1^, (**b**) galvanostatic charge–discharge curves at various current densities, and (**c**) cyclic stability at 1.0 A·g^−1^ over 1000 cycles for materials.

**Table 1 polymers-13-03344-t001:** N 1*s* data, intrinsic oxidation state, and doping degree of conducting materials from XPS results.

Materials	Binding Energy (eV)				
	N1s			[=N–]/[N–H]	[N^+^]/[=N– + N–H]
PANI@TiC	398.51	399.71	400.83	0.56	0.05
PANI@ TiO_2_	398.39	399.53	400.33	0.29	0.51
PANI@Al_2_O_3_	398.22	399.48	//	0.42	//
PANI	398.24	399.52	400.99	0.22	0.60
Assignments	=N−	−NH−/−NC−	−N^+^−/=N^+^−		

**Table 2 polymers-13-03344-t002:** Absorption band, redox peak, and bandgap energy (*E_g_*) of the synthesized materials.

Materials	Absorption Band (nm)		Redox Peaks (V)						*E_g_* (eV)
	**π–π***	**n–π***	* **E** _ **ox1** _ *	* **E** _ **red1** _ *	∆ * **E** _ **p1** _ *	* **E** _ **ox2** _ *	* **E** _ **red2** _ *	∆ * **E** _ **p2** _ *	
PANI	314	591	0.40	0.28	0.12	0.93	0.84	0.09	3.22
PANI@Al_2_O_3_	380	606	0.48	0.32	0.16	0.86	0.74	0.12	2.64
PANI@TiC	394	603	0.44	0.25	0.19	0.75	0.59	0.25	3.20
PANI@TiO_2_	316	616	0.38	0.24	0.06	0.69	0.56	0.13	2.62

## Data Availability

The data created in this study are fully depicted in the article.

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
