# Peer review of "Development, Investigation, and Comparative Study of the Effects of Various Metal Oxides on Optical Electrochemical Properties Using a Doped PANI Matrix"

_polymers, 2021, doi:10.3390/polym13193344_

Round 1

Reviewer 1 Report

The abstract is quite confused and many mistakes are present. The authors should revise carefully it.

The introduction must be written again; English must be revised and relevant literature must be reported, with a focus on what is useful for this work. The authors should also highlight the novelty of the work, which is not evident.

2.1 materials should be reported in a more clear way

2.2 with "chemical preparation" do the authors mean synthesis?

Moreover, if the synthesis is already known, the corresponding reference should be reported.

Fig. 2 How the authors did the peaks deconvolution? it does not always appear consistent.

3.2 can the author better developed the spectra description? The peaks arising from the PANI-MO interaction give information about the nature of the bond and the structure.

Similar comments are valid for all the sections.

This work must be completely rewritten and the results should be analyzed, presented, and discussed furtherly. It might be reconsidered after major revision.

Author Response

Dear reviewer, the detailed answers can be found in the attachment. 

With best regards

Reviewer 2 Report

The manuscript submitted by Amina Bekhoukh et al reported on the “Comparative study of nanocomposites based on conducting polymer matrix with metals oxides using in-situ chemical polymerization method” In this work, three nanocomposites were given, and they were characterized by XPS, XRD, TGA, UV-Vis, FTIR, TEM and CV. But some of the key issues have to be addressed and the manuscript need to be modified before further consideration.

  1. These three nanocomposites (PANI@Al2O3, PANI@TiC, and PANI@TiO2) or highly similar materials were already synthesized and well-characterized in the past by different research groups. I can list several references here.

Zhang, Dawei, Jian Li, and Jianbin Zheng. "Synthesis and electrochemical properties of PANI–TiC nanocomposite and its electrocatalytic behavior." Materials Letters 93 (2013): 99-102.

Zhang, Liuxue, Peng Liu, and Zhixing Su. "Preparation of PANI–TiO2 nanocomposites and their solid-phase photocatalytic degradation." Polymer degradation and stability 91.9 (2006): 2213-2219.

Zhu, Jiahua, et al. "Electrical and dielectric properties of polyaniline–Al2O3 nanocomposites derived from various Al2O3 nanostructures." Journal of Materials Chemistry 21.11 (2011): 3952-3959.

However, I don’t see any other data or discussion in your manuscript except those characterization, please address your novelty in this paper.

  1. In the title of this paper, the authors mentioned this concept “using in-situ chemical polymerization method”. I assume it should be one of the points you want to highlight. But I don’t see comments or discussion about this in result or discussion sections, so why do you want to emphasize it in the title? If it is not a highly creative method that you want to show the readers, I have the same question again. Where is the novelty in this paper?

  1. I suggest the authors to find at least one application or usage for your polymer materials. The authors mentioned “conducting polymer matrix”, maybe try to test the conductivity. Only listing the characterization of polymers are not enough for the publication especially when those nanocomposites or similar structures are known materials and have been studied widely.

  1. Figure 7 is overlapping with the sentences above it. Please fix it.

Author Response

(The authors gave the same response as above.)

Reviewer 3 Report

Evaluation Report

In this article the authors report on the comparative study of nanocomposites based on conducting polymer matrix with metals oxides using in-situ chemical polymerization method. Synthesis and characterization of PANI@Al2O3, PANI@TiC and PANI@TiO2 nanocomposites have been reported recently by various groups for different applications. But the authors of this manuscript didn’t emphasize on the novelty of their work when compared to the already reported work by stating comprehensive problem statement.

In previously reported articles electrochemical behavior of pristine PANI showed similar results as reported by the authors here in this paper. But the authors failed to explain the impact of metal oxides on the electrochemical behavior of polyaniline. 2nd from the CVs no conclusion can be obtained regarding material application.

A lot of typo and grammatical mistakes are present in the article. Major English revision is required.

Additional work is required to check the application of these materials in energy storage devices by carrying out CVs at different scan rates, Galvanostatic charge discharge and electrochemical impedance spectroscopy.

Author Response

(The authors gave the same response as above.)

Round 2

Reviewer 1 Report

I thank the authors for their answers and modifications. The article might be published in present form.

Author Response

Dear Reviewer,

the authors would like to thank you very much for your time and your valuable comments and suggestions, which helped us to improve our work.

Yours sincerely

Dr. Lilia Sabanitna

Reviewer 2 Report

The authors make significant and complete modification based on the comments. I believe it can be accepted.

Author Response

Dear Reviewer,

the authors would like to thank you very much for your time and your valuable comments and suggestions, which helped us to improve our work to a more scientific level.

Yours sincerely

Dr. Lilia Sabanitna

Reviewer 3 Report

The resubmitted version of this manuscript still needs major revision. This judgment is based  on the following comments.

Abstract

  1. The authors write in the abstract that the materials are suitable for solar cell application. However, one cannot find such study in the manuscript.
  2. The very first sentence of the introduction is not right. Conducting polymers cannot make copolymers with metal oxides such as Al2o3 and TiC etc.
  • The authors write that conductive polyaniline (PANI) due to its properties such as high conductivity, broad absorbance, simple synthesis, ease of coupling with metal ions, porous morphology structure, environmental stability as well as ease of production can be  used in numerous applications such as  supercapacitors, electro-chromic devices,  DSSC solar cells,  light emitting diodes and anti-corrosion coatings. However, surprisingly the authors don’t like to give reference of these fascinating properties of polyaniline from “ polymers” and other journals but rather incorporate references on polyaniline/TiO2 and PANI/ PEO etc from other journals. For guidance please read  Electrochimica acta 320 (2019) 134544 ,  Polymers 12(2020) 2870, polymers 12(2020) 2705, Polymers 12(2020) 2212, Polymers 2021,  ,13, 2329, Polymers 13 ( 2021)  2329, Materials chemistry and physics 273, ( 2021) 125071 , Colloids and Surfaces A: Physicochemical and Engineering Aspects 626 (2021) 127076 and several others.

Materials and methods

  1. The authors write that Ammonia solution (NH4OH) (Merck, 25 %) were used for the fabrication of nanocomposites. I think replace fabrication with preparation.

Correct the sentence “The X-Ray photoelectron spectroscopy (XPS) spectra with AVG-Microtech-Multilab 3000-electron spectrometer was using to obtain XPS spectra”

Results and Discussion

  1. Correct the sentence “The XPS survey spectra for all nanocomposites showed contribution from N with the latter being due to PANI (Figure 1)”.
  2. With respect to Figure 1 and 2 it would be more appropriate to include spectra of pristine PANI in order to validate the author’s claims in the same pattern as in FTIR , XRD and UV-visible spectra shown in Figure 3,4 and 5.
  • With respect to UV-Visible spectra the authors write “In the nanocomposites, the intensity of the bands π-π* and  n-π* transitions decreases in comparison to PANI, and these bands are shifted to lower wavelengths. They reduce the extended conjugated system of PANI by decreasing the degree of orbital overlap of electrons and phenyl ring with a single nitrogen atom”. These observations are depicted in figure 5 a.  These observations are in contrast to the bandgap values presented in Table 2. The shifting of  n-π* transitions to lower wavelength as compared to PANI literally means decrease in conductivity and hence bandgap.
  • The TGA spectra presented in Figure 6 needs further elaboration and discussion (for guidance please read TGA section in polymers 13 (2021) 2883.

  1. Sizes of the nanocomposites should be reported based on TEM analysis.
  2. Electrochemical stability is an important aspect of electrochemical materials for application in energy storage devices. It is advised to study electrochemical stability of these materials from CV and GCD curves.
  3. The authors write “The results indicate that the PANI@TiO2 nanocomposite is more suitable for high-performance supercapattery application because of its higher electrical conductivity and lower band gap compared to PANI”. However, one cannot find supercapattery study in this manuscript.
  • Conclusion needs to be revised in the light of revised manuscript.
  • English needs major revision.

Author Response

(The authors gave the same response as above.)
